# Prognostic Value of Late Gadolinium Enhancement in Left Ventricular Noncompaction: A Multicenter Study

**DOI:** 10.3390/diagnostics12102457

**Published:** 2022-10-11

**Authors:** Wei Huang, Ran Sun, Wenbin Liu, Rong Xu, Ziqi Zhou, Wei Bai, Ruilai Hou, Huayan Xu, Yingkun Guo, Li Yu, Lu Ye

**Affiliations:** 1Department of Radiology, Key Laboratory of Birth Defects and Related Diseases of Women and Children of Ministry of Education, West China Second University Hospital, Sichuan University, Chengdu 610017, China; 2Department of Radiology, Hospital of Chengdu University of Traditional Chinese Medicine, Chengdu 610075, China; 3Department of Pediatric Cardiology, West China Second University Hospital, Sichuan University, Chengdu 610017, China; 4Department of Ultrasound, West China Second University Hospital, Sichuan University, Chengdu 610017, China

**Keywords:** cardiovascular magnetic resonance, left ventricular noncompaction, hypertrabeculation, diagnostic criteria, risk stratification, late gadolinium enhancement

## Abstract

Current diagnostic criteria for left ventricular noncompaction (LVNC) may be poorly related to adverse prognosis. Late gadolinium enhancement (LGE) is a predictor of major adverse cardiovascular events (MACE), but risk stratification of LGE in patients with LVNC remains unclear. We retrospectively analyzed the clinical and cardiovascular magnetic resonance (CMR) data of 75 patients from three institutes and examined the correlation between different LGE types and MACE based on the extent, pattern (including a specific ring-like pattern), and locations of LGE in LVNC. A total of 51 patients (68%) presented LGE. A specific ring-like pattern was observed in 9 (12%). MACE occurred in 29 (38.7%) at 4.3 years of follow-up (interquartile range: 2.1–5.7 years). The adjusted hazard ratio (HR) for patients with ring-like LGE were 6.10 (95% CI, 1.39–26.75, *p* < 0.05). Free-wall or mid-wall LGE was associated with an increased risk of MACE after adjustment (HR 2.85, 95% CI, 1.31–6.21; HR 4.35, 95% CI, 1.23–15.37, respectively, *p* < 0.05). The risk of MACE in LVNC significantly increased when the LGE extent was greater than 7.5% and ring-like, multiple segments, and free-wall LGE were associated with MACE. These results suggest the value of LGE risk stratification in patients with LVNC.

## 1. Introduction

Left ventricular noncompaction (LVNC), an uncommon cardiomyopathy characterized by a thickened endocardial layer with prominent trabeculae and a thinned, compacted epicardial layer [1], can occur as an isolated anomaly or associated with left ventricular dilation or hypertrophy, or various forms of congenital heart disease [2,3,4], and may lead to serious outcomes such as heart failure, thromboembolism, implantable cardioverter-defibrillator (ICD) therapy, heart transplantation, or sudden cardiac death [5,6]. Although several definitions have been proposed, currently diagnosis is mainly based on morphologic findings, and have the risk of overdiagnosis [1,7,8,9]. Several studies show a poor correlation between diagnostic criteria and adverse clinical events [10,11,12]. Furthermore, risk stratification in LVNC is particularly challenging and not available. A recent JACC paper proposes a risk prediction model of LVNC, in which late gadolinium enhancement (LGE) being one of the main prognostic factors [13,14]. LGE has negative prognostic implications in heart diseases, including hypertrophic cardiomyopathy, dilated cardiomyopathy, and LVNC [12,15,16,17,18,19,20]; however, previous studies have focused primarily on the presence of LGE but do not provide a detailed risk stratification analysis. The relationship between dose-response, LGE location and pattern, and specific clinical outcomes are poorly understood [14]. Therefore, we conducted a multicenter study to evaluate the association between the extent, location, or pattern of LGE and the impact on prognosis in patients with LVNC and hypertrabeculation patients. It is of great significance for clinicians and radiologists to judge the risk stratification of LVNC and hypertrabeculation patients intuitively and concisely. The guidelines of LVNC were lacking currently, and this evaluation may contribute to the establishment of hypertrabeculated LVNC guidelines and direct clinical management strategies (Figure 1).

## 2. Materials and Methods

### 2.1. Study Population

We searched the clinical and cardiovascular magnetic resonance (CMR) databases at three institutions for patients’ diagnosis descripted as LVNC or hypertrabeculation between January 2013 and December 2018, and the total number is 85. The CMR images of these patients were measured again according to Petersen’s criteria by two radiologists with at least two years of experience. The inclusion criteria followed Petersen et al.’s CMR criteria: (1) CMR images with a distinct two-layered appearance of trabeculated and compacted myocardium; (2) There are prominent myocardial trabeculations in the noncompacted myocardium and deep intertrabecular recesses communicating with the LV; and (3) Subjects with increased LV trabeculation as measured by a noncompaction/compacted (NC/C) ratio ≥ 1.0 anywhere in the myocardial segments on the CMR images. Current diagnosis is mainly based on this criterion [8,14]. Both LVNC and hypertrabeculation patients were included in the cohort. The exclusion criteria were as follows: (1) presence of other known coexisting cardiac abnormalities, including congenital heart disease, coronary heart disease, valvular heart disease, hypertrophic cardiomyopathy, dilated cardiomyopathy or other types of cardiomyopathy, and myocarditis; (2) non-enhanced or poor image quality; and (3) incomplete clinical records. A total of 75 patients were eventually included in the cohort. The Ethics Committee of Clinical Trials and Biomedicine at the West China Hospital of Sichuan University and the ethics committees of other authors’ hospitals approved this study. This study was performed in accordance with the Declaration of Helsinki (2000). Informed consent was obtained from all participants before study participation.

### 2.2. CMR Protocol

Standard gadolinium-enhanced CMR scanning was performed by a 3.0-T whole-body scanner (Skyra; Siemens Medical Solutions, Erlangen, Germany) with a two-element cardiac phased-array coil and ECG-triggering device. After acquiring localization images, 8–12 continuous short-axis cine images that covered the whole left ventricle were obtained by steady-state free-precession sequence. All cine images were acquired at end-expiration. Gadobenate dimeglumine (MultiHance 0.5 mmol/mL; Bracco, Milan, Italy) was injected intravenously at a dose of 0.1–0.2 mL/kg body weight and a flow rate of 2.5–3.0 mL/s after a 20–25 mL saline flush at a rate of 3.0 mL/s. LGE images were obtained using the inversion recovery TrueFISP sequence 10–15 min after contrast injection.

### 2.3. Image Analysis

All CMR data were analyzed using the commercially available software cvi42 (Circle Cardiovascular Imaging, Inc., Calgary, AB, Canada). Image analysis was performed to evaluate conventional cardiac function. For the cine imaging analysis, left ventricular structure and function parameters were measured on the short and long axis at end-diastole and end-systole, respectively. The left ventricular geometric parameters included the ratio of non-compacted to compacted myocardium, the end-diastolic volume (EDV), and end-systolic volume (ESV) mass. Cardiac function measurements, including the LV EDV, LV ESV, LV ejection fraction (EF), LV stroke volume, and LV mass, were analyzed by manually tracing the endocardial and epicardial contours. When delineated, the papillary muscles were excluded from the compacted myocardium. The end-diastolic and end-systolic phases were defined as those with maximum and minimum visual areas, respectively. The endocardial and epicardial boundaries of all images were manually delineated by a radiologist with at least two years of experience who was blinded to the clinical information. In addition, the ratio of NC/C was calculated in end-diastole. In each of three diastolic long-axis views (i.e., horizontal and vertical long-axis and LV outflow tract), the segment with the most pronounced trabeculations was chosen for measurement of the thickness of the non-compacted and the compacted myocardium perpendicular to the compacted myocardium, and only the maximal ratio was then used for analysis. The presence and extent of LGE were assessed and quantified on short-axis images, and LGE was deemed present if myocardial enhancement was confirmed on short-axis areas by using a signal intensity threshold of 5 standard deviations (SD) [21,22] above the mean signal of the remote normal myocardium, expressed as a percentage of scar mass/total LV mass. The presence of LGE was determined by two independent radiologists, with a third providing adjudication if necessary. An experienced radiologist categorized the LGE location and pattern. Visual assessment for the presence and distribution of LGE areas for each left ventricular (LV) myocardial segment was done using a standard 17-segment cardiac model. The definition of ring-like was that LGE present in at least three contiguous segments in the same short-axis slice [23].

### 2.4. Follow-Up

All patients were followed up on the telephone by using the standard questionnaire interview and the clinical medical records, which was performed by experienced physicians blinded to the clinical and CMR data. The clinical endpoint of this study was MACEs, defined as HF hospitalizations, thromboembolic events, appropriate ICD therapy, heart transplant, and sudden cardiac death. The duration of follow-up was calculated from the date of first CMR examination to the first occurrence of an endpoint. The median follow-up was 4.3 years (interquartile range: 2.1–5.7 years).

### 2.5. Statistical Analysis

Statistical analysis was performed with SPSS software (version 26.0, IBM Corp., Armonk, NY, USA). Continuous variables are expressed as means ± SD and categorical data as percentages. Baseline characteristics were compared using the Kruskal–Wallis rank test for continuous data and the Fisher exact test for categorical data. The Kaplan–Meier method was used to evaluate survival, and multivariate Cox regression analyses were performed to compute the hazard ratio (HR) and 95% confidence interval (CI). A *p*-value < 0.05 was significant. The proportional hazard models were adjusted for LVEF, sex, and age, which may confound the association between LGE and results.

## 3. Results

### 3.1. Study Population

The final cohort comprised 75 patients; a total of 52 (69.3%) were men, the median LVEF was 29.1% (IQ range: 17.5–37.7%), and LGE was present in 51 (68%). Patients with and without LGE had similar baseline ages. Patients with LGE had higher diastolic blood pressures (*p* = 0.023), lower LVEF (*p* = 0.005), greater LVEDV (*p* = 0.001) and LVESV (*p* < 0.001), larger left ventricular systolic mass (*p* = 0.001), and diastolic mass (*p* = 0.001). NYHA class ≥ III (*n* = 38, 50.7%) were common in all participants, and patients with LGE tended to have a poor NYHA functional (*p* = 0.01). Baseline characteristics are presented in Table 1. Two experienced radiologists determined LGE, and there was no significant difference between the two diagnoses.

Of the patients with LGE, 9 (12%) patients displayed a ring-like pattern and 42 (56%) a non-ring-like scar pattern. LGE was present only in the septum in 27 (36%) patients, only in the LV free wall in 16 (21.3%), and in both locations in 8 (10.7%). LGE was present only in a single segment in 37 (49.3%) patients, and in multiple segments in 14 (18.7%). LGE was categorized as mid-wall in 43 (57.3%) patients, and non-mid-wall in 8 (10.7%). Additionally, the LGE extent was categorized as three groups ( >0 and ≤7.5%, >7.5% and ≤15%, and >15%).

### 3.2. Outcome of Follow-Up

Over a median follow-up period of 4.3 years (IQ range: 2.1–5.7 years), two patients were lost. Of the 73 patients who completed follow-up, MACEs occurred in 29 (38.7%) patients. A total of 9 (31.0%) had HF hospitalizations, 8 (27.6%) underwent primary prevention ICD implantation, 2 (6.9%) had cardiac transplantation, and 10 (34.5%) had sudden cardiac death. MACEs occurred in 25 (50%) patients with LGE and in 4 (17.4%) without LGE (HR: 5.39; 95% CI: 1.59–18.31; *p* = 0.007). After adjustment of LVEF, age, and sex, LGE was associated with a higher risk of MACEs (HR: 3.84; 95% CI: 1.10–13.40; *p* = 0.035, Table 2 and Figure 2A).

Extent of LGE. Estimated adjusted HRs for patients with LGE extents of 0–7.5%, 7.5–15%, and >15% were 2.01 (95% CI: 0.50–8.03; *p* = 0.323), 7.42 (95% CI: 1.76–31.3; *p* = 0.006), and 9.02 (95% CI: 2.11–38.52; *p* = 0.003), respectively, compared to the patients without LGE (Table 2 and Figure 2B).

Pattern of LGE. Estimated adjusted HRs for patients with ring-like and non-ring-like scar were 6.10 (95% CI: 1.39–26.75; *p* = 0.016) and 3.59 (95% CI: 0.99–12.39; *p* = 0.053) compared to those patients without LGE. Patients with LGE only in single segment and in multiple segments had adjusted HRs for the MACEs of 2.96 (95% CI: 0.82–10.69; *p* = 0.098) and 8.35 (95% CI: 2.10–33.17; *p* = 0.003) compared to those patients without LGE (Table 3 and Figure 3A). 

Location of LGE. Estimated adjusted HRs for patients with LGE only in the septum, only in the free-wall, and in both locations were 2.57 (95% CI: 0.69–9.60; *p* = 0.16), 4.92 (95% CI: 1.18–20.58; *p* = 0.029), and 10.29 (95% CI: 2.42–43.75; *p* = 0.002), respectively, compared to the patients without LGE. Patients with free-wall LGE had an estimated adjusted HR of 2.85 (95% CI: 1.31–6.21; *p* = 0.008) compared to those without free-wall LGE. Patients with mid-wall LGE and other myocardial locations had adjusted HRs for the MACEs of 4.35 (95% CI: 1.23–15.37; *p* = 0.023) and 1.86 (95% CI: 0.30–11.61; *p* = 0.507), respectively, compared to those without LGE (Table 3 and Figure 3B).

NC/C ratio. The NC/C ratio was categorized as hypertrabeculation (NC/C ratio ≥1 and <2.3) and non-compaction (NC/C ratio ≥ 2.3); the risk between the two groups did not reach statistical significance (HR: 1.03 95% CI, 0.49–2.17; *p* = 0.93) (Table 4).

## 4. Discussion

The major clinical messages arising from our study were as follows:(1)The >7.5% LGE extent may be associated with a significantly poor long-term prognosis in hypertrabeculation and LVNC patients.(2)Ring-like LGE and multiple segments LGE were associated with a particularly high risk of MACEs, which deserves extra clinical attention.(3)The NC/C ratio poorly correlates with clinical outcomes, LGE should be considered in diagnoses as a risk predictor, and our study provided useful risk stratification.

Myocardial fibrosis can significantly affect patients’ prognoses. LGE is of great significance in identifying high-risk patients. This finding has been widely confirmed in heart diseases [24,25]. However, the detail risk stratification about LGE and specific LGE pattern in the prognosis of patients with myocardial hypertrabeculation and LVNC are not yet clear [20]. Seventy-five patients from three study centers with CMR-confirmed hypertrabeculation and LVNC were enrolled. During the median follow-up of 4.3 years (IQ range: 2.1–5.7 years), major cardiovascular events (cardiac death, heart failure, thromboembolism, appropriate ICD therapy, and cardiac transplantation) were endpoints. The risk stratification of LGE extent, pattern, and location was determined. The results showed that LGE had a significant impact on the prognosis of patients with hypertrabeculation or LVNC. Risk of MACEs increased significantly with a greater extent of LGE. Our data also showed that patients with different LGE locations or patterns had a different MACE risk, which facilitates the use of CMR for prognostic risk stratification in hypertrabeculation or LVNC patients with LGE.

Patients with LGE were placed in three groups according to the extent of enhancement. After being adjusted for LVEF, sex, and age, the absence of LGE was still associated with a lower risk of MACE. The risks increased significantly when the LGE extent was greater than 7.5%; HRs were 7.42 and 9.02 in the 7.5–15% and >15% groups, respectively. In a prior study, a ≥15% LGE extent was considered a potentially clinically relevant threshold in HCM [26]; however, our data indicate that the prognosis relevant threshold may be lower in patients with LVNC.

A recent study points out that a specific ring-like LGE pattern is associated with a particularly high risk of malignant arrhythmic events, which are significant and independent of the total LGE burden and the presence of other additional risk factors; the HR of a ring-like pattern in this study was 68.98 (95% CI, 14.67–324.39; *p* < 0.01) compared to the absence of LGE [27]. In our study, the LGE pattern was classified as a ring-like and non-ring-like scar. The definition of ring-like was that LGE was present in at least three contiguous segments in the same short-axis slice (Figure 4) [23]. The result of our study is in line with previous reports. The risk of MACE was significantly higher in ring-like LGE patients than patients with non-ring-like LGE and without LGE by Kaplan–Meier analysis. After multivariate adjustment, the presence of ring-like LGE remained associated with an increased risk of the endpoint (HR: 6.10 95% CI, 1.39–26.75; *p* = 0.016). This finding indicates that ring-like LGE is also a predictor of adverse events in patients with LVNC or hypertrabeculation. In particular, ring-like LGE also presented even the extent of LGE was low (<7.5%). The worse prognosis may relate to insults of the conduction system. In this regard, qualitative indicators are more convenient for radiologists to diagnose than quantitative analyses by CVi.

Analogously, it is more intuitive to determine whether LGE is present in multiple segments of the left ventricle. We showed that patients with multiple LGE segments were at higher risk of MACEs. In contrast, those with single-segment LGE were at similar risk to those without LGE. In patients with multiple LGE segments, LGE is not only present in consecutive segments at the same short-axis level but also scattered in a total of 17 segments.

Mahrholdt et al. showed that in a setting of HHV6 and combined PVB19/HHV6 myocarditis, LGE is predominantly located in the anteroseptal region [28]. Aquaro et al. found that LGE present in the anteroseptal wall in patients with acute myocarditis was associated with a worse prognosis [17]. This finding is similar in DCM in those patients with septal LGE had a higher risk of all-cause mortality [15]. However, we showed that free-wall LGE was associated with increased MACEs in LVNC. A greater risk was seen in concomitant free-wall and septum LGE, which indicates that the prognosis of patients with LVNC or hypertrabeculation may have greater relevance with free-wall LGE. In agreement with previous studies, we observed that the most common distribution of LGE in LVNC or hypertrabeculation patients was mid-myocardial and associated with a poor prognosis, similar to other cardiac diseases [29].

Current criteria for the diagnosis of LVNC lead to high disease prevalence in patients referred for CMR, and the NC/C ratio > 2.3 is common in a large population-based cohort, indicating that the NC/C ratio alone for LVNC or hypertrabeculation diagnosis may have low specificity [10,30]. Some studies suggest that the NC/C ratio and the extent of trabeculation do not correlate with adverse outcomes [31], which is in line with our data. These results suggest that a more comprehensive criteria model including LGE should be used. Our study provides several risk stratification models of LGE that are significantly associated with the prognosis in patients with LVNC.

Study limitation. This study has some limitations. Although this is a multicenter study, the low number of patients limited statistical power. Secondly, all the three centers are large referral hospitals. Some patients who came to our hospitals were referred by multiple primary hospitals with more severe symptoms, therefore, they had a lower LVEF overall, and there may be some selection bias. However, this study also provides more prognostic information for patients with relatively severe symptoms.

## 5. Conclusions

In LVNC or hypertrabeculation, the risk of MACE increases significantly when the LGE extent is greater than 7.5% or presence as ring-like LGE. Moreover, multiple segments and free-wall LGE are associated with MACE. The detailed study and risk stratification of LGE in LVNC or hypertrabeculation patients will help improve the diagnostic criteria and make this criterion more closely to clinical prognosis. This study provided useful models based on the extent, pattern, and location of LGE, which provide a much-needed approach to quantify risk.

## Figures and Tables

**Figure 1 diagnostics-12-02457-f001:**
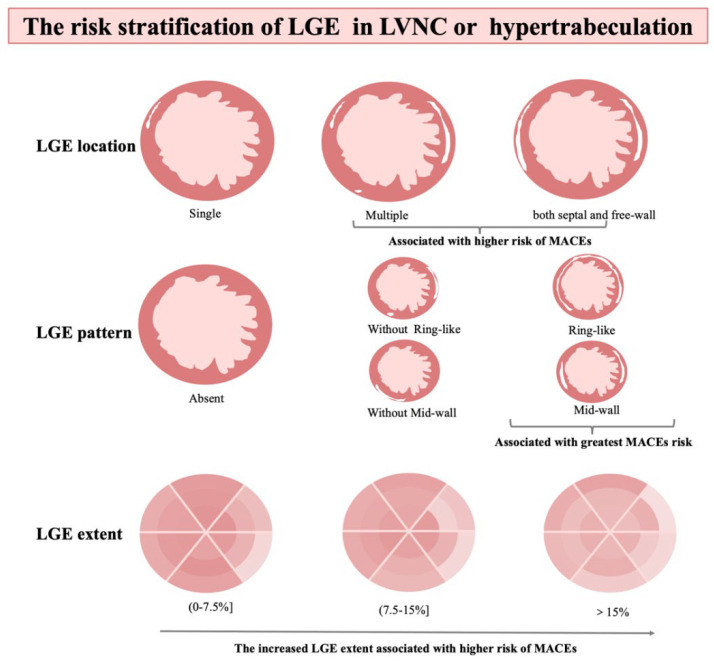
The graphic abstract. CMR, cardiovascular magnetic resonance; NC/C noncompact–tion/compacted; LGE, late gadolinium enhancement.

**Figure 2 diagnostics-12-02457-f002:**
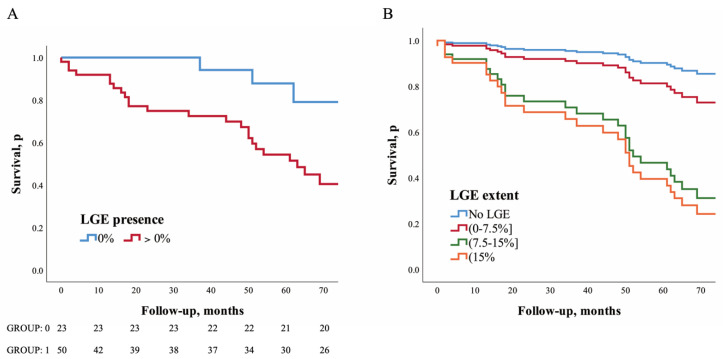
Impact of LGE presence and extent of left ventricular on long–term outcomes. Kaplan–Meier and COX regression analysis survival curves depicting time to MACE. Kaplan–Meier curve for LGE presence (**A**) and Cox regression analysis for LGE extent (**B**); MACE, major adverse cardi–ovascular events.

**Figure 3 diagnostics-12-02457-f003:**
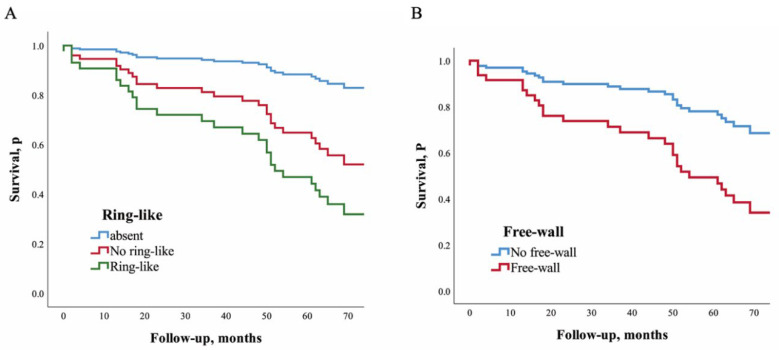
Impact of ring–like and free–wall LGE of left ventricular on long–term outcomes. Cox re–gression analysis for ring–like LGE (**A**) and free–wall LGE (**B**).

**Figure 4 diagnostics-12-02457-f004:**
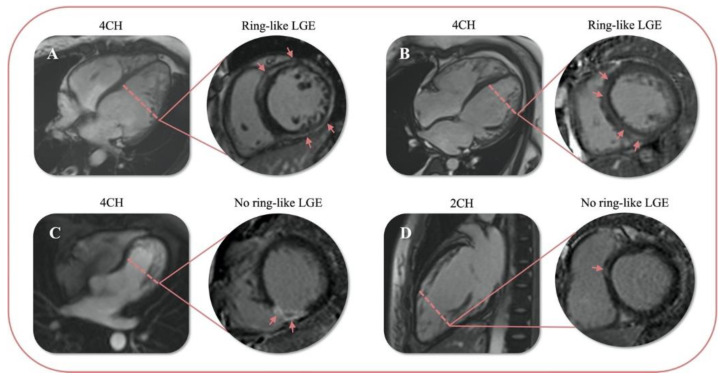
Examples of 4 patients with ring-like or no ring-like LGE. Ring-like LGE (**A**,**B**) and no ring-like LGE (**C**,**D**); 4CH: the four-cavity heart; 2CH: the two-cavity heart.

**Table 1 diagnostics-12-02457-t001:** Baseline Characteristic.

	LGE				
	NO LGE (*n* = 24)	0–7.5 (*n* = 23)	7.5–15 (*n* = 10)	>15 (*n* = 18)	*p* Value *
Age	42.9 ± 20.8	49.0 ± 14.0	50.8 ± 17.7	42.5 ± 14.2	0.382
Male	14 (58.3%)	19 (82.6%)	7 (70%)	12 (66.7%)	0.159
Height	161.9 ± 8.8	165.7 ± 6.1	167.3 ± 6.8	165.8 ± 7.8	0.035
Weight	60.4 ± 12.5	67.7 ± 11.6	67.5 ± 12.0	62.4 ± 9.0	0.108
Heart rate	88.8 ± 21.7	81.8 ± 20.2	77.0 ± 8.2	87.6 ± 31.7	0.185
Systolic blood pressure	112.3 ± 15.5	121.56 ± 16.3	119.7 ± 17.9	113.5 ± 18.1	0.178
Diastolic blood pressure	71.4 ± 13.0	78.6 ± 11.4	77.7 ± 8.8	75.2 ± 12.8	0.023
Hypertension	2 (8.3%)	3 (13.0%)	0	3 (16.7%)	0.656
Alcohol	5 (20.8%)	9 (39.1%)	2 (20%)	7 (38.9%)	0.208
Smoke	5 (20.8%)	11 (47.8%)	1 (10%)	7 (38.9%)	0.158
Medications					
ACE inhibitor	13 (54.2%)	15 (65.2%)	4 (40%)	7 (38.9%)	0.798
Beta-blocker	14 (58.3%)	22 (95.7%)	8 (80%)	15 (83.3%)	0.003
ARB	4 (16.7%)	5 (21.7%)	4 (40%)	6 (33.3%)	0.240
NYHA class ≥ 3	7 (29.2%)	14 (60.9%)	5 (50%)	12 (66.7%)	0.011
CMR measurements					
LVEF (%)	35.2 ± 13.4	25.3 ± 12.6	28.0 ± 14.5	26.2 ± 15.7	0.005
LVEDV (mL)	209.4 ± 65.1	297.4 ± 100.5	289.1 ± 89.0	258.3 ± 85.9	0.001
LVEDVi (mL/m^2^)	120.7 ± 32.3	161.5 ± 52.2	155.4 ± 39.0	145.2 ± 45.5	0.002
LVESV (mL)	140.5 ± 66.8	227.6 ± 89.6	216.7 ± 94.3	200.4 ± 87.3	0.0005
LVESVi (mL/m^2^)	80.8 ± 35.4	123.3 ± 46.2	116.1 ± 45.3	112.5 ± 47.0	0.0009
LVSV (mL)	68.9 ± 25.9	69.8 ± 33.3	72.4 ± 22.4	61.9 ± 21.2	0.883
LVSVi (mL/m^2^)	39.9 ± 13.9	38.2 ± 18.2	39.3 ± 11.8	35.2 ± 12.9	0.388
LV Mass, ED (g)	105.8 ± 33.8	161.3 ± 44.5	127.4 ± 32.5	128.3 ± 43.8	0.001
LV Mass index, ED (g/m^2^)	60.8 ± 16.7	87.3 ± 21.9	68.5 ± 14.1	71.9 ± 22.8	0.001
LV Mass, ES (g)	116.6 ± 37.2	174.9 ± 44.6	142.6 ± 34.2	145.6 ± 54.5	0.001
LV Mass index, ES (g/m^2^)	67.2 ± 19.4	94.8 ± 22.2	76.7 ± 13.9	81.4 ± 28.2	0.001
NC/C	3.0 ± 1.4	2.6 ± 1.6	2.5 ± 1.0	3.0 ± 2.0	0.116

Values are mean ± standard deviation or number (percentage). * Kruskal-Wallis Rank Test for continuous variables; Fisher Exact Test for categorical variables. ACE, angiotensin-converting enzyme; ARB, angiotensin II receptor blocker; NYHA, New York Heart Association; LVEF, left ventricular ejection fraction; LVEDV, left ventricular end diastolic volume; LVESV, left ventricular end systolic volume; LVSV, left ventricular stroke volume; NC/C, non-compacted/compacted ratio.

**Table 2 diagnostics-12-02457-t002:** Individual proportional hazard models investigating the association between major adverse cardiovascular events and late gadolinium enhancement (Presence and Extent).

				Adjusted for LVEF, Sex, and Age		
		*n*	Endpoint	HR (95% CI)	Individual *p* Value	Overall *p* Value
Presence and extent						
LGE	No	23	4 (17.4%)	1.00	-	0.002
	Any	50	25 (50.0%)	3.84 (1.10–13.40)	0.035	
LGE	No	23	4 (17.4%)	1.00	-	<0.001
	(0–7.5%)	23	8 (34.8%)	2.01 (0.50–8.03)	0.323	
	(7.5–15%)	10	7 (70%)	7.42 (1.76–31.3)	0.006	
	>15%	17	10 (58.8%)	9.02 (2.11–38.52)	0.003	

Values are *n* or *n* (%) unless otherwise indicated. *p* values are quoted for each model overall and for the individual components. LVEF, left ventricular ejection fraction; HR, hazard ratio; CI, confidence interval.

**Table 3 diagnostics-12-02457-t003:** Individual proportional hazard models investigating the association between major ad–verse cardiovascular events and late gadolinium enhancement (Location and Pattern).

				Adjusted for LVEF, Sex, and Age		
		*n*	Endpoint	HR (95% CI)	Individual *p* Value	Overall *p* Value
Location and pattern						
LGE (ring-like)	Absent	23	4 (17.4%)	1.00	-	0.004
	No ring-like	41	20 (48.8%)	3.59 (0.99–12.39)	0.053	
	Ring-like	9	5 (55.6%)	6.10 (1.39–26.75)	0.016	
LGE (segment)	Absent	23	4 (17.4%)	1.00	-	0.001
	Single	36	16 (44.4%)	2.96 (0.82–10.69)	0.098	
	Multiple	14	9 (64.3%)	8.35 (2.10–33.17)	0.003	
LGE (Free-wall)	No	51	17 (33.3%)	1.00		0.002
	Yes	22	12 (54.5%)	2.85 (1.31–6.21)	0.008	
LGE (Free-wall)	Absent	23	4 (17.4%)	1.00	-	0.001
	Septal only	27	12 (44.4%)	2.57 (0.69–9.60)	0.160	
	Free-wall only	15	7 (46.7%)	4.92 (1.18–20.58)	0.029	
	Both	8	6 (75%)	10.29 (2.42–43.75)	0.002	
LGE (Mid-wall)	Absent	23	4 (17.4%)	1.00	-	
	Without Mid-wall	8	2 (25%)	1.86 (0.30–11.61)	0.507	
	Mid-wall	42	23 (54.8%)	4.35 (1.23–15.37)	0.023	

Values are *n* or *n* (%) unless otherwise indicated. *p* values are quoted for each model overall and for the individual components. LVEF, left ventricular ejection fraction; HR, hazard ratio; CI, confidence interval.

**Table 4 diagnostics-12-02457-t004:** Individual proportional hazard models investigating the association between major adverse cardiovascular events and NC/C ratio.

NC/C	Adjusted for LVEF, Sex, and Age				
	*n*	Endpoint	HR (95% CI)	Individual *p* Value	Overall *p* Value
Myocardial hypertrabeculation(NC/C ration ≥ 1 and <2.3)	29	13 (44.8%)	1.00	-	0.93
Noncompaction(NC/C ratio ≥ 2.3)	44	16 (36.4%)	1.03 (0.49–2.17)	0.93	

Values are *n* or *n* (%) unless otherwise indicated. *p* values are quoted for each model overall and for the individual components. NC/C, noncompaction/compacted; HR, hazard ratio; CI, confidence interval.

## Data Availability

Not applicable.

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
