# Peer review of "Prognostic Value of Late Gadolinium Enhancement in Left Ventricular Noncompaction: A Multicenter Study"

_diagnostics, 2022, doi:10.3390/diagnostics12102457_

Round 1

Reviewer 1 Report

I found the purpose of the paper relevant and there is a need  for such analysis in this very specific and rare group of patients. Unfortunately, the study group is inhomogeneous.  The authors included in the study patients with both hypertrabeculation who actually didn’t meet the  LVNC criteria and LVNC patients who meet the criteria for diagnosis proposed by Petersen.

According Petersen’s  at. al “The diastolic ratio of >2.3 showed high diagnostic accuracy for distinguishing pathological LVNC from the degrees of non-compaction observed in healthy, dilated, and hypertrophied hearts”. So in fact the results of Wei Huang and colleagues article applies to both patients with LVNC but also DCM, hypertension, HCM and other conditions. On the other hand, it’s all known, that both the presence and location of LGE in DCM and HCM increase the risk of MACE. And these in fact could altered the results.

The authors did not give us the size of the hypertrabeculation group and the LVNC group. I propose that the authors revise the article, will present the results separately for the hypertrabeculation group and separately for the LVNC.

The authors should provide us also with more detailed information regarding trabeculation measurement

-          Was it measured at end diastolic/ end systolic, in the long/short axis

-          Was  the apical segment (17) included or excluded for NC/C measurement,

-          Was only one segment NC/C ≥1 sufficient to recognize hypertrabeculation?, if no, how many?

-          Was only one segment NC/C ≥2,3 sufficient to recognize LVNC, if no, how many?

Have they excluded other causes for LGE such as for example  myocarditis?

Author Response

1) I found the purpose of the paper relevant and there is a need  for such analysis in this very specific and rare group of patients. Unfortunately, the study group is inhomogeneous.  The authors included in the study patients with both hypertrabeculation who actually didn’t meet the  LVNC criteria and LVNC patients who meet the criteria for diagnosis proposed by Petersen.

According Petersen’s  at. al “The diastolic ratio of >2.3 showed high diagnostic accuracy for distinguishing pathological LVNC from the degrees of non-compaction observed in healthy, dilated, and hypertrophied hearts”. So in fact the results of Wei Huang and colleagues article applies to both patients with LVNC but also DCM, hypertension, HCM and other conditions. On the other hand, it’s all known, that both the presence and location of LGE in DCM and HCM increase the risk of MACE. And these in fact could altered the results.

The authors did not give us the size of the hypertrabeculation group and the LVNC group. I propose that the authors revise the article, will present the results separately for the hypertrabeculation group and separately for the LVNC.

Response: 

Thank you for your suggestion.

Considering the trabecular increase may be congenital or acquired, and in order to explore whether the extent of the trabecular affect prognosis, we put the NC/C standard to 1.0. We found the degree of trabecular has no relationship with the prognosis (in the NC/C part of result, table 4). Therefore, the degree of trabecular hyperplasia may not be a critical prognostic factor, so we did not use the NC/C ratio for subgroups.

Meanwhile, we excluded the presence of other known coexisting cardiac abnormalities, including congenital heart disease, coronary heart disease, valvular heart disease, hypertrophic cardiomyopathy, dilated cardiomyopathy or other types of cardiomyopathies, and myocarditis. I am sorry that we did not write in detail in this part. We have added and updated the methods part.

2) The authors should provide us also with more detailed information regarding trabeculation measurement

-          Was it measured at end diastolic/ end systolic, in the long/short axis

-          Was  the apical segment (17) included or excluded for NC/C measurement,

-          Was only one segment NC/C ≥1 sufficient to recognize hypertrabeculation?, if no, how many?

-          Was only one segment NC/C ≥2,3 sufficient to recognize LVNC, if no, how many?

Have they excluded other causes for LGE such as for example  myocarditis?

Response: 

Thank you very much for your suggestion, which really help us a lot to improve our manuscript.

By the standard of Petersen et al., the ratio of NC/C was calculated in end-diastole. In each of three diastolic long-axis views (i.e., horizontal and vertical long-axis and LV outflow tract), the segment with the most pronounced trabeculations was chosen for measurement of the thickness of the non-compacted and the compacted myocardium perpendicular to the compacted myocardium, and only the maximal ratio was then used for analysis. The diagnostic criteria do not involve the number of NC segments; therefore, we did not discuss the number of NC segments separately in the Methods to avoid confusion.

we excluded the presence of other known coexisting cardiac abnormalities, including congenital heart disease, coronary heart disease, valvular heart disease, hypertrophic cardiomyopathy, dilated cardiomyopathy or other types of cardiomyopathies, and myocarditis.

Reviewer 2 Report

Interesting and very useful material for clinical practice. I think that introduction may be improved by reference to daily routine practice implication.

Congrats

Author Response

I think that introduction may be improved by reference to daily routine practice implication.

Response: 

Thank you very much for your suggestion, which really help us a lot to improve our manuscript. We have revised the introduction according to your suggestion.